# HeadsetOff: Enabling Photorealistic Video Conferencing on Economical VR Headsets

Yili Jin
McGill University
yili.jin@mail.mcgill.ca

Xize Duan
The Chinese University of Hong Kong, Shenzhen
xizeduan@link.cuhk.edu.cn

Fangxin Wang
The Chinese University of Hong Kong, Shenzhen
wangfangxin@cuhk.edu.cn

Xue Liu
McGill University
xue.liu@mcgill.ca

## Abstract

Virtual Reality (VR) headsets have become increasingly popular for remote collaboration, but video conferencing poses challenges when the user's face is covered by the headset. Existing solutions have limitations in terms of accessibility. In this paper, we propose `HeadsetOff`, a novel system that achieves photorealistic video conferencing on economical VR headsets by leveraging voice-driven face reconstruction. `HeadsetOff` consists of three main components: a multimodal attention-based predictor, a generator, and an adaptive controller. The predictor effectively predicts user future behavior based on different modalities. The generator employs voice input, head motion, and eye blink to animate the human face. The adaptive controller dynamically selects the appropriate generator model based on the trade-off between video quality and delay, aiming to maximize Quality of Experience while minimizing latency. Experimental results demonstrate the effectiveness of `HeadsetOff` in achieving high-quality, low-latency video conferencing on economical VR headsets.

## CCS Concepts

• **Information systems** → Multimedia streaming; • **Human-centered computing** → Virtual reality.

## Keywords

Video Conferencing, Virtual Reality, VR Headset

**ACM Reference Format:**
Yili Jin, Xize Duan, Fangxin Wang, and Xue Liu. 2024. HeadsetOff: Enabling Photorealistic Video Conferencing on Economical VR Headsets. In *Proceedings of the 32nd ACM International Conference on Multimedia (MM '24), October 28-November 1, 2024, Melbourne, VIC, Australia.* ACM, New York, NY, USA, 9 pages. https://doi.org/10.1145/3664647.3681432

## 1 Introduction

In the past decade, Virtual Reality (VR) has become increasingly popular. With the growing popularity of VR headsets, such as the

Apple Vision Pro [3], Meta Quest [28], and HTC VIVE [16], these devices are no longer just for immersive entertainment experiences; they are becoming integral tools for remote collaboration. This leap forward presents a wealth of opportunities as well as a unique set of challenges, particularly in the realm of video conferencing, which is a crucial function in remote working. A significant issue arises when a user wears a VR headset, which obscures the upper half of the face. In practical VR applications, users typically lack an external camera to capture the lower half of the face, necessitating reliance solely on the data collected by the headset itself.

To address this problem, two main technical solution routes have emerged: cartoon-style avatars and photorealistic reconstruction. Cartoon-style avatar solutions, such as VRChat [45] and Microsoft Mesh [29], involve creating a cartoon-like avatar that users can control, similar to playing a video game. On the other hand, photorealistic reconstruction solutions, like Apple Persona [4], attempt to reconstruct the real human face hidden under the VR headset, providing an experience akin to traditional 2D video conferencing.

However, photorealistic reconstruction solutions typically demand high-precision hardware, such as those found in the Apple Vision Pro, which features a sophisticated sensor array that includes, but is not limited to, LiDAR, Time-of-Flight (ToF) cameras, and infrared cameras. The device is also equipped with a specially designed R1 chip to process the input from these sensors. As a result, these solutions are often less accessible to the broader public due to the complexity and cost of the required hardware.

So there is an interesting and meaningful task at hand: *can we achieve photorealistic video conferencing on an economical VR headset?* To accomplish this, we must address two main challenges: *1) How can we reconstruct the lower half of the face without using high-precision sensors to capture this information? 2) How can we decrease the delay in processing, given that video conferencing is a delay-sensitive task?*

To tackle the first challenge, inspired by recent developments in voice-driven talking head synthesis, we propose using voice to reconstruct the lower half of the face. By leveraging the user's voice input, we can generate a realistic representation of the mouth and jaw movements, eliminating the need for high-precision sensors. For the second challenge, we propose a two-fold approach. Firstly, we suggest predicting user behavior, including voice and head motion, for the next video chunk. By generating the future video chunk in advance, we can significantly decrease the perceived delay during video conferencing. Secondly, we propose jointly considering network adaptive bitrate with reconstruction. Traditional

approaches generate the highest quality video and then send lower-quality versions based on network conditions. Instead, we recommend preparing several models with different quality levels and choosing the corresponding model based on the current network environment. This approach ensures that the video quality is optimized for the available bandwidth while also reducing the impact of reconstruction delay.

We propose `HeadsetOff`, which consists of three main parts: predictor, generator, and controller. The predictor utilizes a multi-modal attention-based network to predict the user's future behavior based on head motion, eye blink, voice, and gaze direction modalities. The generator employs voice input, head motion, and eye blink to animate the human face, providing a range of models with varying quality levels. The controller uses an adaptive algorithm to select the appropriate generator model based on the trade-off between video quality and delay, aiming to maximize Quality of Experience while minimizing latency by dynamically adjusting the model selection based on the current buffer level and the balance between quality and delay.

Experiments show that our predictor achieves superior performance compared to baseline methods in predicting future user behavior, with the lowest Mean Absolute Error (MAE) and Root Mean Squared Error (RMSE) values. The generator produces high-quality, photorealistic videos by leveraging voice, head motion, and eye blink information, achieving comparable results to state-of-the-art methods in terms of Frechet Inception Distance (FID), Cumulative Probability Blur Detection (CPBD), Cosine Similarity (CSIM), and Lip Sync Evaluation (LSE) metrics. Furthermore, our adaptive controller successfully maximizes the QoE while minimizing latency by dynamically selecting the appropriate generator model based on network conditions, outperforming fixed bitrate and buffer-based approaches in terms of Average Video Quality (AVQ) and Rebuffering Ratio (RR). A user study involving 30 participants confirms the effectiveness of `HeadsetOff` in delivering an immersive and engaging video conferencing experience on economical VR headsets, with participants praising the photorealistic facial reconstruction and responsive animations.

The main contributions of this paper are as follows:

- We propose a novel system, `HeadsetOff`, that achieves photorealistic video conferencing on economical VR headsets by leveraging voice-driven face reconstruction.
- We introduce a multimodal attention-based predictor that effectively predicts user future behavior.
- We present a generator that employs voice input, head motion, and eye blink to animate the human face.
- We develop an adaptive controller that dynamically selects the appropriate generator model based on the trade-off between video quality and delay.

## 2 Related Work

In this section, we present related works on traditional and VR video conferencing, along with talking head synthesis.

### 2.1 Traditional Video Conferencing

Video conferencing has undergone a remarkable transformation since its inception in the 1960s. The early Picturephone [32] paved the way, but high costs hindered widespread adoption until the 1990s when affordable ISDN lines emerged [23]. IP-based solutions in the late 1990s revolutionized the industry, making video conferencing accessible to businesses and consumers alike [10]. Researchers focused on enhancing quality through networking [39, 47] and encoding techniques [25, 40]. The COVID-19 pandemic catalyzed an unprecedented surge in video conferencing usage across various domains [43]. However, the limitations of 2D screens have spurred interest in more immersive solutions like VR video conferencing to bridge the gap between remote and in-person interactions.

### 2.2 VR Video Conferencing

In recent years, the evolution of 360° videos [21, 22] and volumetric videos [17, 18, 20, 26] has significantly heightened interest in VR video conferencing. Both researchers and companies are actively exploring diverse methods to enhance user experiences and tackle the challenges associated with VR headsets, which obscure the user's face. Two main technical solution routes have emerged: cartoon-style avatars and photorealistic reconstruction.

Cartoon-style avatar solutions aim to create a virtual representation of the user using a cartoon-like avatar. VRChat [45] is a prominent example of this approach, allowing users to create and customize their avatars using a variety of tools and assets. The avatars are controlled by the user's movements and gestures, captured through VR controllers and tracking systems. Microsoft Mesh [29] is another platform that utilizes cartoon-style avatars for VR collaboration and communication. It leverages Microsoft's Mixed Reality technology to enable users to interact with virtual objects and environments alongside their avatars. These solutions provide a more immersive and engaging experience compared to traditional video conferencing, but may lack the personal connection and non-verbal cues that face-to-face interactions offer.

Photorealistic reconstruction solutions, on the other hand, focus on reconstructing the user's real face and expressions under the VR headset. Apple Persona [4] is a notable example of this approach, using machine learning techniques to generate a realistic 3D model of the user's face based on information including LiDAR. The system captures the user's facial expressions and eye movements using cameras and sensors mounted on the VR headset, and maps them onto the reconstructed 3D model in real-time. Other researchers have explored similar techniques for photorealistic face reconstruction in VR. Thies et al. [42] proposed a face tracking and reconstruction method that uses an RGB-D camera to capture the user's face and a convolutional neural network to regress the facial parameters. Olszewski et al. [34] developed a system that combines a dynamic texture model with a deep appearance model to generate high-fidelity 3D face renderings in VR.

### 2.3 Talking Head Synthesis

The field of talking head video synthesis and manipulation has seen remarkable progress in recent years. Thies et al. [41] introduced a real-time technique for capturing and reenacting facial expressions from RGB videos, enabling convincing synthesis and manipulation of human face video content.

Following this, researchers focused on enhancing various aspects of talking head synthesis. Zakharov et al. [48] developed a

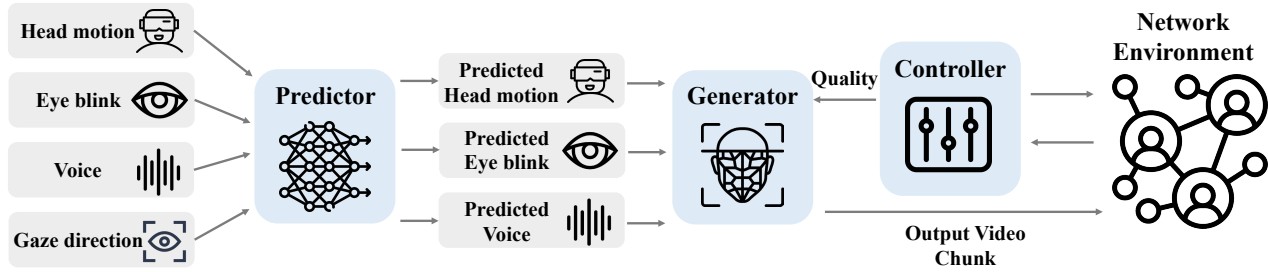

Figure 1: Overview of HeadsetOff.

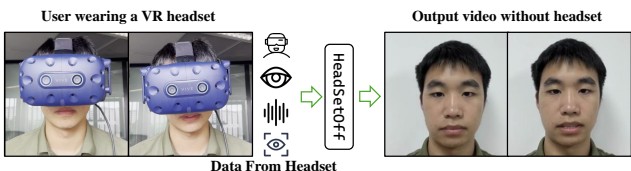

Figure 2: Demo of HeadsetOff.

few-shot learning approach using generative adversarial networks (GANs) [11] to create personalized talking head models from just a handful of frames. Deng et al. [38] proposed a GAN-based method to animate static portrait photos into dynamic videos. Wang et al. [46] introduced a high-quality neural talking head synthesis model with local free-view control that does not require 3D models.

In more recent developments, researchers have harnessed the power of neural radiance fields (NeRF) [30] to achieve highly photorealistic results. Bai et al. [5] utilized a 3D-aware NeRF prior to reconstruct facial avatars from monocular video, demonstrating superior performance in novel view synthesis and face reenactment. Li et al. [24] proposed HiDe-NeRF, a subject-agnostic deformable NeRF that enables high-fidelity free-view talking head synthesis from a single shot.

## 3 System Overview

Figure 1 shows the overview of HeadsetOff, and Figure 2 presents a demo. Our system consists of three main parts: predictor, generator, and controller.

For the predictor, a multimodal attention-based network is utilized to predict the user's future behavior based on head motion (HM), eye blink (EB), voice (VO), and gaze direction (GD) modalities. The predictor aligns the dimensions of different modalities and encodes them using positional and temporal information. The encoded modalities are then processed by Modal Fusion Cores, which capture crossmodal and self-attention to generate fused multimodal vectors for the final prediction.

For the generator, an initialization process is performed to extract facial attribute features, canonical keypoint landmarks, and a 3D Morphable Model (3DMM) from an input frame. During runtime, the generator employs voice input, head motion, and eye blink to animate the human face. The voice input is encoded and mapped to facial expression parameters, which are then integrated with the

canonical keypoints. The face attribute feature is warped based on the updated keypoints, and the resulting representation is fed into a neural network to generate the output video chunk. The generator provides a range of models with varying quality levels to balance Quality of Experience (QoE) and latency.

For the controller, an adaptive algorithm is employed to select the appropriate generator model based on the trade-off between video quality and delay. The controller operates on a set of available video quality levels and maintains a buffer to store generated video chunks. It aims to maximize QoE while minimizing latency by dynamically adjusting the model selection based on the current buffer level and the balance between quality and delay, controlled by a Lagrange multiplier. The controller updates the buffer level, selects the optimal video quality level, and generates the video chunk using the chosen model for each video chunk.

## 4 Predictor

This section presents the predictor's overall structure. The multimodal sequences are initially processed by the Dimension Alignment and Modal Encoding modules. These components are responsible for incorporating positional information and ensuring dimensional compatibility. The processed sequences are then passed to the Modal Fusion Core, which captures multimodal attention. The following subsections provide an in-depth explanation of each module's architecture.

### 4.1 Dimension Alignment and Modal Encoding

Due to the dimensional mismatch between modalities, it is necessary to align them to a common dimension. As the user's behavior tends to exhibit continuity in movement, the temporal information within each modality plays a crucial role in prediction. To emphasize the temporal relationship of neighboring time step information in each modality, we employ a 1-D convolutional network. This process ensures that the different modalities are projected to the same dimension.

The modality sequence possesses temporal characteristics. To enhance the temporal relationship between each time step $t$, we apply positional encoding ($PE$) to the modality sequence. Additionally, we introduce timestamp encoding ($TE$) to further augment the modality sequence and better capture periodic information. The encoded modal sequence can be represented as:

$$E_{HM,EB,VO,GD} = M_{HM,EB,VO,GD} + PE(t) + TE(t). \quad (1)$$

The encoded sequence is then forwarded to the modal fusion core.

## 4.2 Crossmodal Attention

We employ the crossmodal attention mechanism to model the intrinsic relationships between different inputs. This subsection provides a detailed description of this process.

To enhance understanding, let's discuss two distinct modalities: $\alpha$ and $\beta$. The sequence vectors for these modalities are denoted as $M_\alpha$ and $M_\beta$, respectively.

We aim to compute the crossmodal attention $\Phi_{\beta \to \alpha}$ from modality $\beta$ to modality $\alpha$. To facilitate this, we introduce three specific types of weight matrices: $W_{Q_{\beta \to \alpha}}$, $W_{K_{\beta \to \alpha}}$, and $W_{V_{\beta \to \alpha}}$. These matrices project the input modality sequences into distinct representational subspaces.

The Query vector $Q_{\beta \to \alpha}$ is generated by applying the matrix $W_{Q_{\beta \to \alpha}}$ to the sequence $M_\alpha$. This vector encapsulates the feature information of the $\alpha$ modality.

The Key vector $K_{\beta \to \alpha}$ is created by applying the matrix $W_{K_{\beta \to \alpha}}$ to the sequence $M_\beta$. It plays a crucial role in weighting the attention mechanism across the different modalities.

The Value vector $V_{\beta \to \alpha}$ is produced by applying the matrix $W_{V_{\beta \to \alpha}}$ to $M_\beta$. This vector reflects the feature values of the $\beta$ modality, essential for the attention process.

We use the Query vector $Q_{\beta \to \alpha}$, Key vector $K_{\beta \to \alpha}$, and Value vector $V_{\beta \to \alpha}$ to perform the "Scaled Dot-Product." The dot product of the Query vector and the Key vector is divided by $\sqrt{d_k}$. It then passes through a softmax function and transforms into an activated sequence. The product of the activated sequence and the Value vector yields the crossmodal attention score $\Phi_{\beta \to \alpha}$. The detailed calculation formula is as follows:

$$\Phi_{\beta \to \alpha} = softmax(\frac{Q_{\beta \to \alpha} K_{\beta \to \alpha}^T}{\sqrt{d_k}}) V_{\beta \to \alpha}. \quad (2)$$

The value of each entry in $\Phi_{\beta \to \alpha}$ represents the attention score of an entry at a specific position in the $\alpha$ modality sequence to a specific entry in the $\beta$ modality. For example, the $i$-th time step of $\Phi_{\beta \to \alpha}$ is the product result of the Value vector $V_{\beta \to \alpha}$ and the $i$-th row in the weight matrix $softmax(.)$.

## 4.3 Modal Fusion Core

Based on the crossmodal attention mechanism, we create four distinct Modal Fusion Cores for HM, EB, VO, and GD modalities. Each Modal Fusion Core includes crossmodal attention blocks from the three other modalities to the host modality and self-attention blocks based on the standard self-attention mechanism [44]. For instance, the EB modality's Modal Fusion Core calculates three crossmodal attention vectors $\Psi_{HM \to EB}$, $\Psi_{VO \to EB}$, $\Psi_{GD \to EB}$, and a self-attention vector $\Omega_{EB \to EB}$. It then fuses these four vectors into a single fused multimodal vector $F_{EB}$ using a learned weight matrix $W_{concatenate}$.

The crossmodal attention block is composed of $D$ multi-head crossmodal attention layers, representing the attention perspective from multiple dimensions. Each input sequence to the crossmodal attention block passes through a normalization layer. The output vector from the multi-head crossmodal attention layer is processed feed-forwardly by a feed-forward layer. The self-attention block

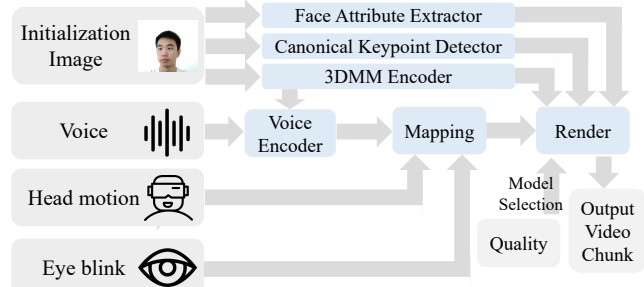

**Figure 3: Overview of Generator.**

resembles the attention mechanism in the Transformer architecture, capturing the attention information within a single modality.

The fused multimodal vectors $F_{HM}$, $F_{EB}$, $F_{VO}$, and $F_{GD}$ obtained from the Modal Fusion Cores are concatenated and passed through a fully connected layer to generate the final prediction.

## 5 Generator

The generator, as illustrated in Fig. 3, commences by receiving an initialization image as input to extract its features. During runtime, it employs voice, head motion, and eye blink to animate the human face. Additionally, we provide a range of models with varying quality levels, enabling the controller to optimize the balance between Quality of Experience (QoE) and latency.

### 5.1 Initialization

To initialize the generator, we first require a photo of the user. This initialization process is performed only once, prior to the video conferencing session. We begin by extracting three distinct feature categories from the input image:

- **Facial Attribute Feature:** The primary feature we strive to extract encompasses the presence of a human face, including characteristics such as skin tone and eye color. This feature facilitates the identification and preservation of the individual's facial attributes in the synthesized view.
- **Canonical Keypoint Landmarks:** These landmarks serve the purpose of representing the unique geometric properties of an individual's face in a neutral pose and expression. By extracting these landmarks, we can model and transform the subject's facial structure in the generated view.
- **3D Morphable Model (3DMM) [6]:** The 3DMM is a statistical model that represents the 3D shape and texture of human faces. By fitting the 3DMM to the input frame, we can estimate the 3D shape, pose, and expression of the face.

These three features will be concurrently extracted by three dedicated networks: Facial Attribute Extractor, Canonical Keypoint Detector, and 3DMM Encoder. For the 3DMM, we leveraged the pretrained model from [8]. The design of the other two networks is described as follows:

**Facial Attribute Extractor:** To enhance the representation of the face and its attributes, we employ an attribute feature extraction network. This network plays a pivotal role in transforming the input frame into an attribute feature representation. The primary

objective of this network is to extract intricate facial features from the input frame. It accomplishes this by utilizing a sequence of downsampling blocks that strategically leverage convolution layers to convert the initial 2D features into a comprehensive representation. Subsequently, a series of residual blocks is employed to meticulously compute the final features, which encapsulate the essence of the frame's three-dimensional structure.

**Canonical Keypoint Detector:** Drawing inspiration from prior works [9, 12, 46], we introduce a canonical keypoint landmark detector to capture the distinctive geometric characteristics of an individual's face in a neutral pose and expression. The detector adopts a U-Net-like [36] encoder-decoder architecture. The encoder employs several convolutional layers to encode the input frame into a latent representation. Notably, rather than projecting the encoded features into a 2D space, we expand them into 3D using a $1 \times 1$ convolutional bottleneck layer. This enables us to extract 3D canonical keypoint landmarks instead of solely 2D landmarks. The decoder mirrors the encoder with a series of upsampling 3D convolutional layers. At each upsampling step, the resolution of the features is increased while incorporating higher-level semantic information from the corresponding encoder layer via skip connections. This facilitates the recovery of the original spatial dimensions for landmark prediction. Finally, the output of the last decoder layer generates the predicted coordinates of the canonical keypoint landmarks.

## 5.2 Voice Encoder and Mapping

As inspired by [55], constructing model capable of generating precise facial expression from voice input presents a formidable challenge due to the inherently complex nature of mapping audio to expressions, which varies considerably across individual identities. So we incorporate 3DMM parameters into the voice encoder.

Once the voice input is encoded into facial expression parameters, we integrate elements of the user's head movements and eye blinks. This comprehensive mapping results in a more nuanced and lifelike representation of facial expressions that are synchronized with the voice input.

## 5.3 Render

After obtaining the aforementioned features, we proceed to generate the final output video chunk through the following steps:

**Canonical Keypoints Fusion:** The adjusted face expression is integrated with the canonical keypoints by concatenating them together. This fusion process allows for the combination of the structural information captured by the keypoints, creating a comprehensive representation that encapsulates both the facial structure and the desired expression. The fused representation combines the canonical keypoints and the adjusted face expression, enabling a holistic representation of the facial characteristics.

**Face Attribute Feature Transformation:** Subsequently, we apply warping to the face attribute feature, utilizing the updated canonical keypoints as a guide. This warping step ensures that the extracted facial attributes are appropriately transformed and aligned with the adjusted face expression, preserving the spatial relationships between the attributes and the facial structure. The warping function, guided by the fused representation, transforms the face attribute feature to align it with the desired facial expression.

**Output Processing:** Finally, the warped face attribute feature, enhanced with the adjusted canonical keypoints, is fed into a neural network. This network utilizes the amalgamated information to produce the output. It is important to note that this step is the most computationally intensive. To address this, we train models with varying quality levels; a high-level model with more parameters can generate high-quality video but with a longer processing delay. The choice of which model to use is determined by the controller.

## 6 Controller

In this section, we present our Controller to select the appropriate generator model based on the trade-off between video quality and delay. The goal is to maximize the Quality of Experience (QoE) for the viewer while minimizing the latency in video generation.

### 6.1 Problem Formulation

The controller aims to optimize the QoE by selecting the most suitable video quality level for each chunk, considering the trade-off between quality and delay. The objective can be expressed as:

$$\max_{i_k \in V} \sum_{k=1}^{K} \left( q_{i_k} - \lambda \cdot \frac{d_{i_k}}{B(t_k)^\gamma} \right), \tag{3}$$

where $i_k$ is the selected video quality level for chunk $k$, $q_{i_k}$ is the corresponding quality score, $d_{i_k}$ is the delay for generating the video at level $i_k$, $B(t_k)$ is the current buffer level at time $t_k$, $\lambda$ is the Lagrange multiplier, and $\gamma$ is a utility function parameter controlling the trade-off between quality and delay.

The objective function is subject to the following constraints:

$$B(t_k) = \min \left( B_{\max}, B(t_{k-1}) + \Delta t - d_{i_{k-1}} \right), \ 0 \le B(t_k) \le B_{\max}, \tag{4}$$

where $B_{\max}$ is the maximum buffer size, and $\Delta t$ is the time elapsed since the last decision.

### 6.2 Algorithm Description

The Controller operates on a set of available video quality levels, each associated with a quality score and a generation delay. As general video conferencing system, it maintains a buffer to store generated video chunks and aims to keep the buffer level close to a maximum threshold. The model selection is adapted based on the current buffer level and the balance between quality and delay, controlled by a Lagrange multiplier. The Controller follows these steps for each video chunk $k$ at time $t_k$:

(1) Calculate the time elapsed since the last decision, $\Delta t$.
(2) Update the current buffer level $B(t_k)$ based on the elapsed time and the delay of the previously selected model. The buffer level is capped at the maximum buffer size $B_{\max}$.
(3) Select the optimal video quality level $i_k$ for chunk $k$ by maximizing the utility function that considers the quality score $q_i$, the delay $d_i$, the current buffer level $B(t_k)$, and the Lagrange multiplier $\lambda$. The utility function parameter $\gamma$ controls the trade-off between quality and delay.
(4) Update the Lagrange multiplier $\lambda$ based on the difference between the current buffer level and the maximum buffer size, scaled by a step size parameter $\beta$.
(5) Update the last decision time $t_{\text{last}}$.

---

**Algorithm 1** Controller

---

1: $V \leftarrow$ Set of available video quality levels
2: $q_i \leftarrow$ Quality score for video level $i \in V$
3: $d_i \leftarrow$ Delay for generating video at level $i \in V$
4: $B_{\max} \leftarrow$ Maximum buffer size
5: $B(t) \leftarrow$ Current buffer level at time $t$
6: $\gamma \leftarrow$ Utility function parameter
7: $\lambda \leftarrow$ Lagrange multiplier
8: $t_{\text{last}} \leftarrow$ Last decision time
9: **for** each video chunk $k$ at time $t_k$ **do**
10: $\quad \Delta t \leftarrow t_k - t_{\text{last}}$
11: $\quad B(t_k) \leftarrow \min \left( B_{\max}, B(t_{\text{last}}) + \Delta t - d_{i_{k-1}} \right)$
12: $\quad i_k \leftarrow argmax_{i \in V} \left( q_i - \lambda \cdot \frac{d_i}{B(t_k)^\gamma} \right)$
13: $\quad \lambda \leftarrow \lambda - \beta \cdot (B(t_k) - B_{\max})$
14: $\quad t_{\text{last}} \leftarrow t_k$
15: $\quad$ Generate video chunk $k$ using model with quality $i_k$
16: **end for**

---

(6) Generate video chunk $k$ using the model with the selected quality level $i_k$.

The Controller provides an adaptive approach to select the appropriate video generation model based on the current buffer level and the balance between video quality and delay. By dynamically adjusting the model selection, the algorithm aims to optimize the QoE for the viewer while managing the latency in video generation. The Lagrange multiplier serves as a control parameter to maintain a stable buffer level close to the maximum threshold.

## 7 Experiment

This section presents experiments conducted with HeadsetOff, including a user study of the entire system and detailed evaluations of each module.

### 7.1 User Study

Quantitatively evaluating the entire HeadsetOff system is challenging. To qualitatively assess the overall Quality of Experience (QoE) provided by HeadsetOff, we conducted a user study involving 30 participants. We randomly divided the participants into 15 pairs and asked them to engage in two video conferencing sessions. In the first session, each pair used a traditional video conferencing system for approximately 3 minutes. Subsequently, they engaged in another 3-minute session using HeadsetOff.

The participants found that HeadsetOff could reconstruct the human face well, providing an experience similar to traditional 2D video conferencing with a direct camera feed. The majority of the participants expressed a positive impression of HeadsetOff, noting that the photorealistic reconstruction of facial features significantly enhanced the sense of presence and immersion. They also appreciated the smooth and lifelike animations of facial expressions driven by voice input, head motion, and eye blinks. Several participants commented that the experience felt engaging and personal, closely resembling face-to-face interactions.

However, a few participants mentioned that in certain instances, the reconstructed faces appeared unnatural or exaggerated. They

suggested that further refinements to the voice-to-expression mapping algorithm could improve the realism of the generated animations, especially for extreme facial expressions or rapid changes in emotion.

Despite these minor limitations, the overall qualitative evaluation indicated that HeadsetOff delivers a highly immersive and engaging video conferencing experience on economical VR headsets. The photorealistic facial reconstruction and responsive animations were identified as the key strengths of the system, contributing to an enhanced sense of presence and enabling more natural communication between participants.

### 7.2 Predictor

To evaluate the performance of our multimodal attention-based predictor, we conducted experiments using a dataset collected from 30 participants. The dataset consists of head motion (HM), eye blink (EB), voice (VO), and gaze direction (GD) sequences recorded during video conferencing sessions.

We compared our predictor with the following baselines:

- LSTM: A long short-term memory (LSTM) network [15] that processes the concatenated multimodal sequences.
- Transformer: A standard Transformer model [44] that uses self-attention to capture the dependencies within the multimodal sequences.

We evaluated the predictors using two metrics: Mean Absolute Error (MAE) and Root Mean Squared Error (RMSE). MAE measures the average absolute difference between the predicted and ground-truth values, while RMSE emphasizes larger errors by taking the square root of the mean squared error. Lower values of MAE and RMSE indicate better prediction performance.

Table 1 presents the prediction results on the test set. Our predictor achieves the lowest MAE and RMSE values among all the compared methods, demonstrating its effectiveness in predicting future user behavior based on multimodal sequences. The LSTM and Transformer baselines perform relatively well but fail to capture the intricate relationships between different modalities. Our predictor's ability to model the interactions between modalities and capture the dependencies within each modality contributes to its superior performance.

These experimental results demonstrate the effectiveness of our multimodal attention-based predictor in capturing the complex relationships between different modalities and predicting future user behavior. The predictor's ability to anticipate user actions enables HeadsetOff to generate future video chunks in advance, significantly reducing the perceived delay during video conferencing.

### 7.3 Generator

To evaluate the performance of our generator, we collected normal talking head videos from video conferences, along with the corresponding voice, head motion, eye blink, and gaze direction data. We used this information as input and the collected videos as ground truth. We then compared the reconstruction accuracy between the generated and original videos. We used several metrics to assess the quality and realism of the generated videos:

- Frechet Inception Distance (FID) [14]: FID measures the similarity between the distributions of generated and real images.

**Table 1: Prediction results. HM: Head Motion, EB: Eye Blink, VO: Voice. Lower values are better.**

| Method | MAE (HM) | MAE (EB) | MAE (VO) | RMSE (HM) | RMSE (EB) | RMSE (VO) |
|---|---|---|---|---|---|---|
| LSTM | 0.183 | 0.092 | 0.164 | 0.241 | 0.119 | 0.213 |
| Transformer | 0.165 | 0.087 | 0.152 | 0.220 | 0.113 | 0.198 |
| **Our Predictor** | **0.141** | **0.076** | **0.138** | **0.189** | **0.098** | **0.181** |

**Table 2: Generator evaluation results. Lower values are better for FID and LSE-D, while higher values are better for CPBD, CSIM, and LSE-C.**

| Method | FID | CPBD | CSIM | LSE-D | LSE-C |
|---|---|---|---|---|---|
| SadTalker | 22.06 | 0.34 | 0.84 | 7.77 | 7.29 |
| Our Generator | 23.43 | 0.21 | 0.79 | 8.18 | 5.14 |

Lower FID values indicate better quality and realism of the generated videos.

- Cumulative Probability Blur Detection (CPBD) [31]: CPBD assesses the sharpness of the generated images. Higher CPBD values indicate sharper and more detailed results.
- Cosine Similarity (CSIM): We used the cosine similarity of identity embeddings extracted from the ArcFace [7] model to evaluate the preservation of the speaker's identity in the generated videos. Higher CSIM values indicate better identity preservation.
- Lip Sync Evaluation (LSE) [35]: We employed the distance score (LSE-D) and confidence score (LSE-C) from the Wav2Lip model to assess the quality of lip synchronization and mouth shape in the generated videos. Lower LSE-D and higher LSE-C values indicate better lip synchronization.

We compared our generator with a state-of-the-art audio-driven talking head synthesis method, SadTalker [55]. Table 2 presents the evaluation results.

Our generator achieves comparable performance to SadTalker, indicating similar levels of realism in the generated videos. It is important to note that our generator utilizes additional input modalities, such as head motion and eye blink, which contribute to more accurate facial expression reconstruction.

These experimental results demonstrate the effectiveness of our generator in producing high-quality, photorealistic videos by leveraging voice, head motion, and eye blink information. The ability to generate videos at different quality levels allows our system to adapt to various network conditions and user preferences, ensuring a smooth and engaging video conferencing experience.

## 7.4 Controller

To evaluate the performance of our controller, we conducted experiments using a network simulator that models a range of network conditions. We utilized the FCC dataset [1], which comprises millions of mobile throughput internet traces.

We compared our controller with the following baselines:

- Fixed Bitrate (FBR): A simple approach that selects a fixed video quality level, regardless of network conditions.

**Table 3: Video adaptation results. AVQ: Average Video Quality (higher is better), RR: Rebuffering Ratio (lower is better).**

| Method | AVQ | RR (%) |
|---|---|---|
| FBR | 2.13 | 8.42 |
| BB | 2.58 | 3.27 |
| **Our Controller** | **3.24** | **1.79** |

**Table 4: Controller performance under different network conditions.**

| Network Condition | AVQ | RR (%) |
|---|---|---|
| Low Bandwidth | 2.41 | 2.85 |
| Medium Bandwidth | 3.12 | 1.93 |
| High Bandwidth | 3.69 | 1.28 |

- Buffer-Based (BB) [19]: A method that selects the video quality level based on the current buffer level, aiming to maintain a stable buffer.

We evaluated the controllers using two metrics: Average Video Quality (AVQ) and Rebuffering Ratio (RR). AVQ measures the average quality level (from 1 to 5) of the generated videos throughout the session, while RR represents the percentage of time spent rebuffering due to insufficient buffer. Higher AVQ and lower RR indicate better performance.

Table 3 presents the evaluation results. Our controller achieves the highest AVQ and lowest RR among the compared methods, demonstrating its effectiveness in adapting to varying network conditions. The FBR approach suffers from frequent rebuffering events, as it cannot adapt to changes in network bandwidth and latency. The BB method maintains a more stable buffer but often selects lower quality levels to avoid rebuffering.

To further analyze the behavior of our controller, we investigated its performance under different network conditions. Table 4 shows the AVQ and RR for three representative network scenarios: low bandwidth (0.5-1.5 Mbps), medium bandwidth (1.5-3.0 Mbps), and high bandwidth (3.0-5.0 Mbps). As expected, the controller selects higher quality levels and achieves lower rebuffering ratios when the network bandwidth is higher. In the low bandwidth scenario, the controller prioritizes maintaining a stable buffer by selecting lower quality levels, resulting in a slightly higher RR. These results demonstrate the controller's ability to adapt to different network conditions and make appropriate quality level decisions.

These experimental results demonstrate the effectiveness of our controller in maximizing the QoE while minimizing latency in video

generation. By dynamically adapting the video quality level based on network conditions and the trade-off between quality and delay, the controller ensures a smooth and engaging video conferencing experience across various network scenarios.

## 8 Discussion

The advancements presented in this paper have significant implications for the accessibility and adoption of immersive remote collaboration solutions. By eliminating the need for expensive, high-precision hardware, `HeadsetOff` lowers the barrier to entry for photorealistic VR video conferencing. This could lead to wider adoption across various domains, such as business meetings, remote education, and social interactions. The enhanced sense of presence and engagement provided by `HeadsetOff` has the potential to greatly improve the quality of remote communication and collaboration.

However, there are several limitations and opportunities for future work that should be acknowledged. While the user study indicated a generally positive reception of `HeadsetOff`, some participants noted instances of slightly unnatural or exaggerated facial expressions. Further refinements to the voice-to-expression mapping algorithm [2] could improve the realism of the generated animations, especially for extreme or rapidly changing emotions. Incorporating additional modalities, such as electromyogram (EMG) signals [33], may provide a richer input representation for more accurate expression synthesis.

Another limitation is the inability to capture silent, subtle movements of the face with economical headsets due to the lack of sensors. This gap underscores an opportunity for future research. Developing methods to accurately record these nuanced expressions could dramatically improve the realism and expressiveness of generated animations. Enhancing cheap sensor or devising new software algorithms to better interpret limited data could be potential approaches to address this issue.

The personalization of the `HeadsetOff` to individual users is another area for future exploration. Currently, the generator is trained on a diverse dataset to handle a wide range of face types and expressions. However, adapting the models to specific users through fine-tuning or few-shot learning techniques could potentially enhance the realism and consistency of the generated animations. Personalized models [27, 52, 53] may better capture the unique facial characteristics and mannerisms of each user, further improving the overall video conferencing experience.

The scalability and robustness of `HeadsetOff` in real-world scenarios also warrant further investigation. While the experiments were conducted in controlled settings, the system should be tested under diverse network conditions [49, 50, 54] and with a larger user base [37, 51] to assess its performance and identify potential challenges. Strategies for handling network fluctuations, packet loss, and other impairments should be developed to ensure a seamless user experience.

Lastly, the ethical implications of photorealistic face animation should be carefully considered [13]. While `HeadsetOff` aims to enhance remote communication, it is crucial to establish guidelines and safeguards to prevent misuse or deception. Users should be made aware of the capabilities and limitations of the system, and their consent should be obtained for the use of their likeness in virtual environments. Additionally, given the risk of misunderstandings caused by exaggerated, missing, or incorrect facial expressions, it is clear that any commercial system must undergo a more systematic evaluation. This evaluation should include a systematic analysis of user preferences for different options, recognizing that perfect accuracy may not always be achievable. Ongoing research and public discourse on the responsible development and deployment of such technologies are necessary to address these concerns and ensure the ethical use of photorealistic VR video conferencing.

## 9 Conclusion

In this paper, we presented `HeadsetOff`, a novel system that achieves photorealistic video conferencing on economical VR headsets by leveraging voice-driven face reconstruction. Our system comprises three main components: a multimodal attention-based predictor, a generator that employs voice input, head motion, and eye blink to animate the human face, and an adaptive controller that dynamically selects the appropriate generator model based on the trade-off between video quality and delay. Through extensive experiments, we demonstrated the effectiveness of `HeadsetOff` in generating high-quality, photorealistic video conferencing experiences while minimizing latency. Our contributions pave the way for more accessible and immersive remote collaboration solutions, enabling seamless interactions in virtual environments without the need for expensive, high-precision hardware.

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
