# OpenReview forum: "HeadSetOff: Enabling Photorealistic Video Conferencing on Economical VR Headsets"
_acmmm.org/ACMMM/2024/Conference — MM2024 Oral_

### Official Review · Reviewer_9577 · 2024-05-05

**Rating:** 4
**Confidence:** 3

**Summary:**

This paper proposed a deep learning-based photorealistic human face animation framework for video conferencing on VR headsets. The framework first predicts the user’s future facial behavior based on their head motion, eye blink, gaze direction and voice. Then a generator animates the mouth and jaw movements of the picture of the user’s face. Finally, a controller algorithm selects the optimal version of the generator to maximize the quality of the human face rendering while minimizing the video conference latency.

**Strengths:**

-the author designed a multimodal face reconstruction scheme that considers head motion, eye motion and voice to vividly animate the user’s jaw and mouth movement.

-the author also tackled the latency issue of transmitting the reconstructed face through video conferencing, making the HeadSetOff a practical framework.

-the adaptive controller dynamically selects the appropriate generator model based on network conditions, ensuring optimal video quality and reduced delay.

-the system's multimodal attention-based predictor effectively anticipates user behavior, enhancing the realism and responsiveness of the animation.

-experimental results show the system achieves high-quality, low-latency video conferencing on economical VR headsets, making advanced VR communication more accessible.

**Limitations:**

-this framework mainly covers jaw and mouth movements that come with sound (e.g., talking). There are movements that are silent, such as smiling. Without recognizing these movements, the reconstructed face is less vivid.

-a prerequisite for the system is a photo of the user’s face, and the quality of this photo (e.g., angle, luminance) can affect animation performance. The evaluation would be more comprehensive if it included the impact of these photo variations.

-the system might produce slightly unnatural or exaggerated facial expressions in some instances, particularly with extreme or rapidly changing emotions.

-personalization to individual users is limited; fine-tuning or few-shot learning techniques could enhance the realism and consistency of the animations by better capturing unique facial characteristics.

**Suitability:**

2

---

### Official Review · Reviewer_arZM · 2024-05-20

**Rating:** 4
**Confidence:** 2

**Summary:**

This paper proposes a videoconferencing method that can be applied to economical VR headsets, which has some advantages over existing mature but expensive products. In the paper, the overall system design is described in detail and sufficient experiments are also carried out to demonstrate the superiority of the proposed system.

**Strengths:**

In this paper, HeadSetoff is proposed as a feasible videoconferencing method via an economical VR headset. The method fully integrates multimodal information and is also expected to further promote the application of VR technology in daily life. The overall system composition is clear and well-organized. The experiments in this work are also relatively sufficient to demonstrate the superiority of the system.

**Limitations:**

Despite the novelty of this paper, there are still some confusions, which I hope the author can answer:

1) Through the first two paragraphs of the introduction, I feel that the difficulty of VR videoconferencing is reconstructing the upper half of the face, due to the occlusion of the VR headset. And around L80 it is mentioned that the difficulty is reconstructing the lower half of the face. Is this a contradiction?
2) According to 1), combined with Fig. 3 and related content, I think there is no distinction between the upper and lower half of the face in Generator, but it is directly driven by the character picture and multimodal information.
3) Average Video Quality (AVQ) is mentioned at L795, but I missed the specific Video Quality Assessment Methods. May I know whether PSNR, SSIM, VMAF or other video quality assessment algorithms are used here?
4) According to my experience, Sadtalker in Table 3, although it is faster in face driver, it is far from real-time. Can Generator in the article do the driving more quickly to enable video conferencing in real-time?
5) Based on the experimental setup in Sec. 7.1, I did not experience a significant superiority of video conferencing using VR headsets compared to traditional 2D video conferencing. Since the text does not explicitly mention whether the final presentation is in 2D or 3D, I divide it into 5-a) and 5-b) to discuss them separately.
5-a) If the Generator's driving method is a 2D driving method and does not provide a free point of view for 3D content, how does the advantage of videoconferencing via a VR headset manifest itself when participating in a 2D videoconference in a bare-eye situation still achieves the same, if not better, results?
5-b) If the Generator's driving method is a 3D driving method, which can have a certain range of viewpoints, then when the character's head moves (e.g., turning the head from left to right, looking up and down), it is obvious that the character image used for initialization cannot cover the full range, and therefore the distortion generated during the synthesis needs to be further discussed.

Overall, I think 5) is the most important of these. Although the authors may have described it in Sec 2.2, I still hope that the authors can provide relevant screenshots or effects to make the readers feel more advanced.

**Suitability:**

3

---

### Official Review · Reviewer_dqU2 · 2024-05-22

**Rating:** 6
**Confidence:** 3

**Summary:**

In this paper, the authors present a new system for photorealistic VR communication, while wearing an HMD.
Fot that they introduced a system that is split up into 3 major parts, a predictor, a generator and a controler.
Every single part is described in depth, in that paper so that it is clear what they are talking about.
The big goal is a reconstruction of a human face while the person is wearing an HMD, so that the other end of the vr video-conferencing call can still see the face and emotions.

**Strengths:**

This paper is very detailed and describes all the important parts in depth, so that the reader should be able to follow the progress of the paper. Each part (predictor, generator, and controller) is described and evaluated individually to show that the authors thought about every possible point, where errors could occure.

Furthermore, a qualitative test was performed and evaluated in this paper as well, to show that the HeadSetOff also works with real Users and not only theoretically.

The authors also discuss the limitations and future steps, which shows, that they know, that there is still work that needs to be done.

**Limitations:**

I would have liked some additional pictures of the subjective test, or atleast a reconstruction of a face, so that I could at least imagine what to expect, when using HeadSetOff.

**Suitability:**

3

---

### Official Review · Reviewer_CMMq · 2024-06-04

**Rating:** 3
**Confidence:** 3

**Summary:**

The paper presents HeadSetOff, a novel system that achieves photorealistic video conferencing on economical VR headsets by leveraging voice-driven face reconstruction. Experiments also demonstrate the effectiveness of HeadSetOff in generating high-quality, photorealistic video conferencing experiences while minimizing latency.

**Strengths:**

This paper intends to deal with a practical issue in VR, which is caused by the fact that the user’s face is covered by the headset. I believe the paper holds potential, and the problem it addresses is relevant to MM. Several experiments were conducted to evaluate the HeadSetOff system.

**Limitations:**

Overall, some critical information is missing, and the rationality of the compared baseline model needs to be further explained and improved. My detailed comments are as follows,

1. This paper devotes a considerable length to discussing related work about video conferencing, while providing insufficient discussion about “Talking Head Synthesis” (Section 2.3), which is more relevant to this paper.

2. Figure 2 illustrates the overview of the proposed predictor. However, it is more like a simple diagram that can not demonstrate the details of the model structure. Moreover, figure 3 is in a similar situation.

3. This paper chooses LSTM and Transformer as two baselines, which are pretty old and not convincing. Some new sequence predicting methods should be considered, such as TimesNet and Autoformer. Same as baseline selections in Section 7.3 and Section 7.4.

4. This paper proposes a HeadSetOff system consisting of three main components. However, each individual module appears slightly insufficient.

5. This paper lacks an ablation study to prove the effectiveness of the designs of the predictor and the generator. Besides, it lacks a visual presentation of the head synthesis.

6.  Page 7, Section 7.4. Do the settings of three different bandwidth values have any basis?

Minor comments
1. The fonts of annotations on some of the figures are inconsistent. For example, the “Head motion”, “Eye blink” in Figure 1 and Figure 2.

2. The annotation (“Model Selection”) on the arrows in Figure 3 is out of bounds. Please re-design this figure and make it decent.

3. The layout of the paper needs to be optimized, as there are excessive gaps between some Sections and paragraphs.

4. Some sentences are a bit colloquial. For example, “So there is an interesting and meaningful task at hand” on Page 1, Line 77.

**Suitability:**

2

---

### Meta-Review · Area_Chair_dKPz · 2024-07-03

**Recommendation:** Accept (Oral)
**Confidence:** 4

**Metareview:**

The paper was generally appreciated by the 4 reviewers. The detailed system description (R2) is clear and well written (R3). The paper - and thus overall system described - is considered to hold potential (R1) addressing a practical issue of reconstructing the otherwise invisible face from parsons wearing an HMD. The face reconstruction method is multimodal (R3, R4), considering head and eye motion to animate the user's jaw and mouth movement (R4). As such it is stated that it may contribute to promoting VR technology for daily life (R3). Addressing also low-delay transmission of the reconstructed face - in a network-adaptive manner - is thought to make HeadsOff a practical framework (R4). Here, the attention-based predictor is considered a further positive, predicting user behavior to enhance realism and responsiveness (R4). The several (R1) experiments for evaluating each part (R2) of the system and the qualitative user test are considered as another strong point (all reviewers), showing "high quality low-latency video conferencing on economic VR headsets" (R4).

The authors have provided a detailed and quite convincing rebuttal, indicating to include how they address the points of criticism in the rebuttal also into the camera-ready (CR) paper. The average rating has improved from 4.25 to 4.5 based on the constructive rebuttal, appreciated by 3 of the 4 reviewers.

Criticisms included some missing information on the baseline model (R1). Further, the imbalance of describing SoA on videoconferencing (VC) vs. talking head synthesis, considered the main contribution of the paper (R1). Here, also, some clarification wrt. reconstructing the lower vs. upper parts of the face were mentioned (R3). Constructive considerations are contained in rebuttal and indicated to be included in CR paper.
More details are requested for the figures describing the system, which have been indicated to be provided in the CR version. Further comparisons to baseline models are included in rebuttal, too (R1). While individual components are considered insufficient (R1), the authors highlight the abilities of the overall system, which appears convincing - also not criticized by other reviewers. Regarding the concern that generator and predictor should be evaluated separately in an ablation study (R1), the authors indicate that they carried this out. Here, though, it is not clear whether this new information will be included in the CR paper.
The wish fort further example images was mentioned by several reviewers (R1, R2), and respective examples are indicated in the rebuttal, to be included in the CR paper.
Points of criticism regarding the "Average Video Quality" construct (R3) are addressed in the rebuttal as well, again to be included in a final revision. Concerns wrt. real-time operation (R3) are addressed by the predictor, as outlined in rebuttal. Here, I recommend to the authors to also clarify the descriptions in a possible CR paper version, to avoid this concern.
The very justified question by R3 regarding the possible superiority of the approach in comparison with classical VC is well addressed in the rebuttal, but here, too, I recommend the authors to clarify this also in the CR paper. IMO, the points raised by R3 in point 5 are still valid, considering a 2D vs. 3D scenario, and should be addressed in revision.
Some limitations will remain and may better be even more clearly be presented in the paper. An important one is that of not being able to capture silent (emotional) movements of the face with economical headsets, as no sensors exist for that. Even though somewhat mentioned, in the CR version this may be discussed more clearly as "avenue for future research" as indicated in the rebuttal.
In addition, the considerations about the initial photo quality should be included in the CR paper as well. The authors mention an additional experiment in the rebuttal, which could be included in CR if feasible.
Due to the potential of misunderstandings and the like produced by exaggerated or possibly missing or wrong expressions, it is evident that any commercial system would need to undergo a much more systematic evaluation, which however cannot be expected from such scientific paper. Here, besides the mentioned future pathways, also a systematic analysis of preferences for different options should be considered, especially if perfect accuracy cannot be achieved. This pertains to the general need for evaluation methods of generative AI.

Overall, the paper appears to be a solid contribution, hence I recommend to accept is as oral presentation.